# Producing an Emulsified Meat System by Partially Substituting Pig Fat with Nanoemulsions that Contain Antioxidant Compounds: The Effect on Oxidative Stability, Nutritional Contribution, and Texture Profile

**DOI:** 10.3390/foods8090357

**Published:** 2019-08-23

**Authors:** Isaac Almaráz-Buendia, Adriana Hernández-Escalona, Roberto González-Tenorio, Nestor Santos-Ordoñez, José Jesús Espino-García, Víctor Martínez-Juárez, Martin A. Meza-Nieto, Rafael Germán Campos Montiel

**Affiliations:** Instituto de Ciencias Agropecuarias, Universidad Autónoma del Estado de Hidalgo, Av. Rancho Universitario s/n Km.1 C.P. 43760 Tulancingo, Hgo., Mexico

**Keywords:** xoconostle, phenols, ABTS, DPPH, TBARS

## Abstract

The objective of this study was the incorporation of a water–oil (W/O) nanoemulsion for the partial substitution of pig fats and the addition of antioxidant compounds in an emulsified meat system (EMS). The nanoemulsion was formulated with orange essential oil and cactus acid fruit (xoconostle). The treatments were different percentages (0, 1, 2, 3, 4, and 5%) of the nanoemulsion for the substitution of pig fat in the EMS. The proximal analysis (moisture, protein, fat, and ash), texture profile (hardness, cohesiveness, springiness, and chewiness), phenolic compounds and antioxidant capacity 2, 2-diphenyl-1-picrylhydrazyl (DPPH), 2,2′-Azino-bis-3-ethylbenzothiazoline-6-sulfonic acid (ABTS), and 2-thiobarbituric acid reactive substances (TBARS) were evaluated. All variables showed significant differences (*p* < 0.05). The results for protein, fat, and ash exhibited increments with the addition of the nanoemulsion, and moisture loss was reduced. The profile showed increments in hardness and chewiness. The addition of the nanoemulsion incremented the phenolic compounds and antioxidant capacity (DPPH and ABTS), decreased production of Malonaldehyde, and reduced lipid oxidation. The result of the addition of the nanoemulsion in the EMS is a product with a substantial nutritional contribution, antioxidant capacity, and excellent shelf life.

## 1. Introduction

In the meat industry, fat is very important for emulsified products. Fat is responsible for emulsion stability and water retention capacity, further providing energy, essential fatty acids, and fat soluble vitamins [1]. The fat soon degrades due to oxidation, thereby producing poor sensory characteristics, discoloration, and rancidity [2]; in addition, fat oxidation results in a reduction in shelf life and production of toxic compounds [3].

Emulsified meat products are enhanced with synthetic antioxidant compounds, including butyl-hydroxytoluene, butyl-hydroxyanisole, and t-butyl-hydroxyquinone, among others, to reduce fat oxidation and extend shelf life [4]. These synthetic products have the disadvantages of promoting toxicological, mutagenic, and carcinogenic effects [5,6,7]. One alternative option is the use of natural antioxidants in meat products [2]. 

The cactus acid fruit, xoconostle, from the genus *Opuntia*, contains phenolic compounds, carotenoids, betacyanins, and betalains [8]. These bioactive compounds have shown antioxidant activity [9] and antibacterial activity [10]. Furthermore, the orange essential oils include terpenes, such as D-limonene, that protect fat against oxidizing compounds [11]. In addition, essential oils have been shown to possess antibacterial, antifungal, and antioxidant activities [12,13], so these components can be used as functional ingredients in foods [14,15]. The antioxidant compounds are sensitive to external factors, such as light, temperature, and oxygen. One way to protect these compounds is by using encapsulation, such as in the form of nanoemulsions. This type of encapsulation has the advantage of improving the transportation and controlling the release of active molecules through the biological membrane [16].

Sharma et al. [17] incorporated four types of essential oils (clove, holy basil, cassia, and thyme) in emulsified chicken and demonstrated a reduction in fat oxidation. Wang et al. [3] substituted pig fat with camellia oil gel in sausage and found favorable results, such as reduced fat, lower moisture, and minor values of 2-thiobarbituric acid reactive substances (TBARS).

The objective of this work was the partial substitution of pig fat with water-oil W/O emulsions containing cactus acid fruit (xoconostle) in an emulsified meat system to evaluate the physicochemical characteristics, texture profile, and oxidative stability for 60 days. The objective of this work was to evaluate the effect of the partial substitution of pig fat with W/O emulsions containing cactus acid fruit (xoconostle) in an emulsified meat system on physicochemical characteristics, texture profile, and oxidative stability. 

## 2. Materials and Methods 

### 2.1. Preparation of the Nanoemulsion 

The nanoemulsion was water in oil (W/O). It was prepared according to the methodology of Guler et al. [18] with some modifications. The continuous phase was orange essential oil (Hilmar Ingredients, USA) (70%), the dispersed phase was the cactus acid fruit (xoconostle) (10%), and the surfactant was liquid soya lecithin (Hilmar Ingredients, USA) (20%). All components were stirred using an ultrasonic processor (Ultrasonic Sonics Vibra-Cell, VCX 130, USA). A 6 mm probe was used and 20 intervals (20 s/interval) of sonication and recesses of 10 s were established to obtain the necessary drop size. The ultrasonic processor was used at 80% amplitude at a frequency of 20 kHz, and the mixture was placed in an ice bath to avoid temperature increases during mixing. The droplet size distribution of the nanoemulsion was determined with the dynamic laser light scattering technique using Zetasizer equipment (Nano-ZS2000 Model Malvern Instruments Ltd. Malvern, Worcestershire, United Kingdom) by placing the sample in a glass cell. Five replicates were considered for each formulation [19]. Furthermore, the phenolic content and antioxidant activity (2, 2-diphenyl-1-picrylhydrazyl (DPPH) and 2,2′-Azino-bis-3-ethylbenzothiazoline-6-sulfonic acid (ABTS)) were determined in the nanoemulsions.

### 2.2. Production of the Emulsified Meat System 

The emulsified meat system was carried out according to the method described by Cofrades et al. [20] with some modifications. Formulations with different percentages of animal fat and nanoemulsion were made (Table 1). The minced meat (1 cm^2^), salt, and ice were placed in a cutter (Dito-Sama F 23200 GBR, Aubusson, France) and beaten for two minutes, and then nanoemulsion was added to the mixture and beaten for one minute more. 

The fat was incorporated into the mixture, maintaining a temperature no higher than 16 °C. The mixture was fed through a stuffing device (BG-PRUFRZERT Inc., City of México, México) and injected into 20 mm diameter synthetic cellulose casings (Viscofan Brand Inc., City of México, México). The filled casings were heated to 72 °C for 30 min, and then subjected to thermal shock by being placed in ice. Finally, the meat emulsion in casings were vacuum packed in bags (Zubex Inc., City of México, México) in a sealer (Tor Rey EVD48, City of México, México) and refrigerated at 4 °C.

### 2.3. Proximal Composition 

The proximal analysis was performed according to the official methods of the Association of Official Agricultural Chemists (AOAC) edited by Horwitz [21]. The moisture was calculated by drying a sample in a stove at 100 °C for 8 h (Official Method 925.09), the fat content by the Soxhlet method (Official Method 923.05), the ash percentage was determined by the incineration of the muffle samples at 550° C for 8 h (Official Method 923.03), and the protein content by the Kjeldahl method (Official Method 981.10).

### 2.4. Texture Profile Analysis (TPA) 

These tests were performed according by Cofrades et al. [22] with some modifications. Eight repetitions were performed for each treatment. Cubes of 1 × 1 × 1 centimeters were elaborated and a texturometer (Brookfield CT3 texture analyzer, Brookfield Engineering Laboratories, Inc. Middleboro, MA, USA) was used. The samples were axially compressed to 50% of their original height with a 4.5 kg load cell at a speed of 1 mm/s, with the use of a TA3/1000 probe and a TA-BT-KI table. The parameters measured were hardness, cohesiveness, springiness, and chewiness. The test was performed at room temperature. 

### 2.5. Total Phenols 

The content of total phenols was done following a modified version of the methodology by Singleton et al. [23]. The samples were diluted to 1:10. Then, 0.5 mL of sample was mixed with 2.5 mL of previously diluted (1:10) Folin-Ciocalteau reagent (Sigma-Aldrich, St. Louis, MO, USA) and 2 mL of 7.5% sodium carbonate (Fermont) was added. The mixture was left for 120 min in total darkness. After, the samples were read in a spectrophotometer (JENWAY 6715 Ultraviolet/Visible (UV/V), Staffordshire, UK) at a wavelength of 760 nm. The results were expressed as mg of gallic acid equivalents for 100 g of emulsified meat system with nanoemulsion (EMSN) (GAE/100 g of EMSN).

### 2.6. DPPH

The methodology of Brand-Williams et al. [24] for DPPH test was used with some modifications. Here, 0.0039 g of DPPH (2,2-diphenyl-1-picrylhydrazyl) (Sigma-Aldrich, USA) in 50 mL of 80% methanol (JT Baker, VWR International. Tultitlán, México) was mixed and left for 2 h in the dark, then calibrated at 0.7 ± 0.1 absorbance. Then, 0.5 mL of this mixture was added to 2.5 mL of DPPH solution and left in darkness for 60 min. After, the samples were read at 517 nm in a spectrophotometer (JENWAY 6715 UV/V, UK). The results were expressed as mg of ascorbic acid equivalents for 100 g of emulsified meat system with nanoemulsion (EMSN) (AAE/100 g of EMSN).

### 2.7. ABTS

Here, 7 mM (10 mL) of 2, 2′-Azino-bis-(3-ethylbenzothiazoline-6-sulfonic acid) (ABTS) (Sigma-Aldrich, St. Louis, MO, USA) and 2.45 mM (10 mL) of potassium persulfate were mixed. The mixture was left in complete darkness for 16 h. Next, the mixture was adjusted with 20% ethanol to obtain a value of 0.7 ± 0.1 absorbance. The final solution (3.9 mL) was taken and 100 μL of sample was added. The mixture was read at 734 nm [25]. The results were expressed as mg of ascorbic acid equivalents for 100 g of emulsified meat system with nanoemulsion (EMSN) (AAE/100 g of EMSN).

### 2.8. 2-thiobarbituric acid reactive substances (TBAR)

Lipid oxidation was evaluated according to Wang et al. [26] with some modifications. An extractor solution was prepared containing 7.5 % trichloroacetic acid (Fermont PA Cert, Monterrey, México), 0.1% gallic acid (Fermont PA Cert, Monterrey, México), and 0.1 % EDTA, disodium salt dehydrate (Baker ACS, México). Then, 2.5 g of samples were taken and homogenized with 25 mL of extractor solution in an Ultraturrax T25 (IKA-Werke GmbH & Co. KG) 3000 rpm for 1 min. The homogenate was centrifuged at 6000× g forces at 20 °C for 10 min. The supernatant (2 mL) was mixed with 80 mM (2 mL) of thiobarbituric acid (BP 50067 lllkirch, Strasbourg, France) (TBA). The mixture was incubated at 40 ° C for 90 min and it was read at 532 nm. The TBARS values were interpreted with the calibration curve of 1,1,3,3-tetramethoxypropane (Malonaldehyde) (Sigma-Aldrich, St. Louis, MO, USA) in different concentrations and the results were expressed in milligrams of malonaldehyde (MDA)per kilogram of sample (mg MDA /Kg).

### 2.9. Statistical Analysis

The experimental design was completely random. The results were analyzed by ANOVA, when there were significant differences (*p* < 0.05), comparison of media (Tukey) was used with the statistical program STATGRAPHICS C. XVI Version 16.1.03 (Statgraphics Technologies Inc., The Plains, VA, USA).

## 3. Results and Discussion

### 3.1. Nanoemulsions and Characterization

The drop diameter was 73 ± 6 nm and the Z potential value was −107 mV. Both parameters are characteristic of nanoemulsions [27,28]. Our results are similar to those reported in Gago et al. [29] for nanoemulsions of clove and lemongrass essential oil. The phenolic content was 184.3 mg GAE/100 g, the antioxidant activity from DPPH was 97.76 mg AAE/100 g, and ABTS was 126.3 mg AAE/100g in the nanoemulsions. 

### 3.2. Proximate Composition

Significant differences (*p* < 0.05) were observed in the moisture of the meat emulsion system in the different treatments and times. Treatments with the nanoemulsions demonstrated a reduced loss of moisture (Table 2), and similar results were reported by Sharma et al. [17] in chicken sausage with the addition of different essential oils. The major reason for the retention of water could be that the nanoemulsions contain soy lecithin in the formulation, which was used as an emulsifier [30,31].

Significant differences (*p* < 0.05) were observed in protein between the different treatments and times. The major protein content was observed in the treatments with nanoemulsions (Table 2). Choi et al. [1] found similar results in the substitution of pig fat with vegetable oil in the emulsified meat system. However, Bolger, Brunton, and Monahan [32] did not find significant differences (*p* > 0.05) in protein content in an emulsified product with encapsulated flaxseed oil. The increment in protein could be due to soy lecithin, which contains amino acids.

The EMSN showed a significant increment (*p* < 0.05) in the content of fat after the addition of the nanoemulsion (Table 2). In contrast, Choi et al. [1] found less fat with the addition of vegetable oils in EMS; however, the quality of the lipid provided by the nanoemulsion is better compared to that provided by pig fat. The orange essential oil contains antioxidant compounds, such as D-limonene, according to Chasquibol et al. [11].

The values of ash were between 1.94 and 1.95 in the treatment with EMSN 5% (Table 2). Choi et al. [1] reported similar results (1.72 to 1.97) with the addition of vegetables oils in EMS.

### 3.3. Texture Profile Analysis (TPA)

The nanoemulsion significantly (*p* < 0.05) affected the hardness of the EMSN. Treatment with the 5% nanoemulsion produced the most substantial hardness (Table 3). Similar results were reported by Youssef and Barbut [33] in a meat batter with canola oil. These authors attributed the increase in the hardness to the oil’s smaller globule size and the enhanced interaction between proteins.

The nanoemulsion did not affect the cohesiveness of the EMSN (Table 3). Wang et al. [3] reported the same results after the partial substitution of pig fat with camellia oil gel in sausage. In contrast, Choi et al. [1] observed an increment after the addition of vegetable oils in an EMSN. 

No significant differences (*p* > 0.05) were observed between treatments with respect to springiness (Table 3). The incorporation of flaxseed oil did not affect the springiness of chicken sausage [32]. The EMSN did not show changes in springiness due to the addition of oils or nanoemulsions.

The EMSN exhibited a significant increment (*p* < 0.05) in chewiness after the incorporation of the nanoemulsion (Table 3). These results coincide with those reported by Youssef and Barbut [33], Choi et al. [1], and Bolger et al. [32] with respect to the substitution of fat with vegetable and seed oils in EMSN. The increase in chewiness is related to the protein incorporated within the nanoemulsion.

The effect on shelf life exhibited significant differences (*p* < 0.05) after the substitution of pig fat with the nanoemulsions. The EMSN showed increased hardness and chewiness but reduced cohesiveness and springiness, and these effects could be attributed to the loss of moisture during storage.

### 3.4. Total Phenols and Antioxidant Activity

The contents of phenols were significantly enhanced (*p* < 0.05) by the incorporation of the nanoemulsions (Table 4) because the nanoemulsions contain phenolic compounds from the xoconostle extract. The addition of cherry extract to sausage also increased the content of phenols [34].

The results of the antioxidant activity (DPPH) assays exhibited significant differences (*p* < 0.05) between the treatments. The major activity was found in the treatment EMSN 5%; this activity was about 1.8-fold greater with respect to the EMSN 0% on day 60. The EMSN 0% exhibited antioxidant activity because the meat contains peptides with antioxidant properties, such as carnosine (β-alanyl-L-histidine) [35]. Sharma et al. [17] incorporated essential oils in chicken sausage and found major inhibition of DPPH radicals. The nanoemulsion contains xoconostle extracts and orange essential oil, which contain bioactive compounds, thus resulting in the increment in antioxidant activity (DPPH). The bioactive compounds inhibit free radicals [36,37,38]

The ABTS radical showed the same results as DPPH, with significant differences between the treatments (*p* < 0.05). Again, treatment EMSN 5% showed major antioxidant activity about 2.2-fold greater than the EMSN 0% (Table 4). Isaza et al. [31] found similar results after the incorporation of cherry extract in sausage. The phenols content and antioxidant activity were reduced with a controlled release during storage (Table 4). Again, treatments with nanoemulsions showed the best results.

Lipid oxidation showed significant differences (*p* < 0.05) between the treatments. Treatment EMSN 5% showed about a 2.7-fold reduction in the production of malonaldehyde (MDA) with respect to the EMSN 0%. Šojić et al. [39], Bianchin et al. [40], Erdmann et al. [41], and Ozogul et al. [42] found that the incorporation of essential oils in (their) meat systems reduced the production of malonaldehyde with respect to the control. The incorporation of the nanoemulsions with antioxidant compounds from xoconostle and orange essential oil delayed lipid oxidation, thus extending the shelf life of the EMSN.

## 4. Conclusions

The incorporation of the nanoemulsion in the emulsified meat system improved the nutritional contribution due to the increment in protein, inclusion of essential oils, and reduction in the loss of moisture. The texture profile showed increased hardness and chewiness. The bioactive compounds and antioxidant activities (DPPH and ABTS) incremented after the incorporation of the nanoemulsions, resulting in reduced production of malonaldehyde and minor lipid oxidation. The most favorable treatment was emulsified meat system with nanoemulsion (EMSN) 5%. Thus, the nanoemulsion extended the shelf life of the emulsified meat system.

## Figures and Tables

**Table 1 foods-08-00357-t001:** Formulation of emulsified meat systems with different percentages of nanoemulsion.

Treatments	Meat %	Fat %	Nanoemulsion %	Ice %	Salt %
**0%**	65	20	0	13	2
**EMSN 0%**	65	19	1	13	2
**EMSN 2%**	65	18	2	13	2
**EMSN 3%**	65	17	3	13	2
**EMSN 4%**	65	16	4	13	2
**EMSN 5%**	65	15	5	13	2

Emulsified meat system with nanoemulsion (EMSN).

**Table 2 foods-08-00357-t002:** Proximal composition of the emulsified meat system with nanoemulsion for the parameters of moisture, protein, fat and ash.

	Days	EMSN 0%	EMSN 1%	EMSN 2%	EMSN 3%	EMSN 4%	EMSN 5%
**Moisture**	1	68.58 ± 0.085 ^aC^	68.53 ± 0.007 ^aB^	68.52 ± 0.019 ^aD^	68.59 ± 0.094 ^aB^	68.65 ± 0.093 ^aE^	68.67 ± 0.092 ^aD^
	15	67.88 ± 0.018 ^aC^	67.94 ± 0.479 ^aB^	67.98 ± 0.099 ^aC^	67.92 ± 0.382 ^aB^	67.98 ± 0.013 ^aD^	67.96 ± 0.112 ^aC^
	30	65.14 ± 0.427 ^aB^	65.60 ± 0.107 ^aA^	65.72 ± 0.077 ^aB^	65.89 ± 0.036 ^aA^	66.05 ± 0.022 ^aC^	66.11 ± 0.083 ^aB^
	45	64.46±0.252 ^aAB^	65.10 ± 0.075 ^bA^	65.23 ± 0.001 ^bA^	65.32 ± 0.001 ^bcA^	65.67± 0.009 ^cdB^	65.80 ± 0.003 ^dA^
	60	63.58 ± 0.002 ^aA^	64.95 ± 0.002 ^bA^	65.13 ± 0.010 ^cA^	65.24 ± 0.011 ^dA^	65.40 ± 0.017 ^eA^	65.58 ± 0.009 ^fA^
**Protein**	1	14.89 ± 0.020 ^aA^	15.09 ± 0.008 ^abA^	15.10± 0.024 ^abA^	15.15±0.009 ^abcA^	15.29± 0.146 ^bcA^	15.40 ± 0.103 ^cA^
	15	15.39±0.041 ^aB^	15.47±0.199 ^aAB^	15.53 ± 0.073 ^aB^	15.63 ± 0.024 ^aB^	15.67 ± 0.306 ^aA^	15.83 ± 0.056 ^aB^
	30	15.49± 0.022 ^aBC^	15.54 ± 0.031 ^aB^	15.61 ± 0.089 ^aB^	15.84 ± 0.017 ^bC^	15.87± 0.037 ^bAB^	16.00 ± 0.035 ^bB^
	45	15.55 ± 0.005 ^aC^	16.08 ± 0.066 ^bC^	16.24 ± 0.023 ^cC^	16.36 ± 0.024 ^cdD^	16.41± 0.054 ^dBC^	16.46 ± 0.035 ^dC^
	60	16.15 ± 0.059 ^aD^	16.65 ± 0.003 ^bD^	16.69 ± 0.002 ^bD^	16.84 ± 0.008 ^cE^	16.89 ± 0.008 ^cC^	16.94 ± 0.012 ^cD^
**Fat**	1	8.83 ± 0.058 ^aA^	9.39 ± 0.007 ^bA^	10.26 ± 0.056 ^cA^	10.60 ± 0.077 ^dA^	11.74 ± 0.012 ^eA^	12.23 ± 0.092 ^fA^
	15	9.35 ± 0.186 ^aB^	10.11 ± 0.037 ^bA^	10.87 ± 0.051 ^cA^	11.12 ± 0.070 ^cB^	12.44 ± 0.147 ^aB^	12.61 ± 0.192 ^dA^
	30	13.38 ± 0.024 ^aC^	14.29 ± 0.257 ^abB^	14.43 ± 0.370 ^bB^	14.75 ± 0.026 ^bC^	14.90 ± 0.322 ^bC^	15.32 ± 0.325 ^bB^
	45	13.95 ± 0.061 ^aD^	14.53 ± 0.646 ^abB^	14.59 ±0.087 ^abB^	14.81± 0.008 ^abC^	15.21 ± 0.099 ^bC^	15.29 ± 0.015 ^bB^
	60	14.52 ± 0.008 ^aE^	14.64 ± 0.068 ^abB^	14.78 ± 0.022 ^bB^	14.95 ± 0.011 ^dA^	15.33 ± 0.012 ^dC^	15.69 ± 0.024 ^eB^
**Ash**	1	1.94 ± 0.037 ^aA^	1.94 ± 0.017 ^aA^	1.94 ± 0.001 ^aA^	1.94 ± 0.005 ^aA^	1.95 ± 0.001 ^aA^	1.95 ± 0.006 ^aA^
	15	1.95 ± 0.002 ^aA^	1.96 ± 0.001 ^abAB^	1.96 ± 0.002 ^abcA^	1.97 ± 0.003 ^bcB^	1.97 ± 0.002 ^cB^	1.97 ± 0.002 ^cB^
	30	1.96 ± 0.001 ^aA^	1.96 ± 0.002 ^aAB^	1.96 ± 0.007 ^aB^	1.97 ± 0.001 ^abB^	1.98 ± 0.001 ^abB^	1.98 ± 0.003 ^bB^
	45	1.97 ± 0.007 ^aA^	1.97 ± 0.007 ^aAB^	1.97 ± 0.002 ^aBC^	1.97 ± 0.001 ^aB^	1.98 ± 0.001 ^aB^	1.98 ± 0.003 ^aB^
	60	1.98 ± 0.002 ^aA^	1.98 ± 0.006 ^aB^	1.98 ± 0.001 ^aC^	1.98 ± 0.004 ^aB^	1.98 ± 0.006 ^aB^	1.98 ± 0.003 ^aB^

Emulsified meat system with nanoemulsion (EMSN). The lowercase letters in the superscript indicate significant differences (*p* < 0.05) between treatments (rows), and uppercase letters indicate significant differences in each treatment with respect to time (columns) (*p* < 0.05).

**Table 3 foods-08-00357-t003:** Texture profile analysis (TPA) for the parameters hardness, cohesiveness, springiness, and chewiness in the emulsified meat system with nanoemulsion.

	Days	EMSN 0%	EMSN 1%	EMSN 2%	EMSN 3%	EMSN 4%	EMSN 5%
**Hardness (N)**	1	12.49 ± 0.344 ^bA^	12.38 ± 0.307 ^bA^	12.63 ± 0.302 ^aA^	13.44± 0.358 ^cA^	14.11 ± 0.306 ^dA^	14.57± 0.333 ^dA^
	15	13.68 ± 0.238 ^bB^	12.93 ± 0.357 ^aB^	12.94 ± 0.406 ^aB^	14.52 ± 0.270 ^cB^	15.16 ± 0.254 ^dB^	15.53 ± 0.252 ^dB^
	30	14.40 ± 0.238 ^bC^	13.51 ± 0.336 ^aC^	14.13 ± 0.374 ^aB^	15.43 ± 0.245 ^cC^	16.57 ± 0.332 ^dC^	16.47 ± 0.406 ^dC^
	45	15.12 ± 0.681 ^cD^	13.73 ± 0.165 ^aC^	14.94 ± 0.373 ^bC^	16.43 ± 0.261 ^dD^	17.58 ± 0.313 ^eD^	17.50 ± 0.288 ^eD^
	60	17.76 ± 0.252 ^cE^	14.59 ± 0.318 ^aD^	15.25 ± 0.356 ^bD^	17.58 ± 0.314 ^cE^	18.40 ± 0.264 ^dE^	18.52 ± 0.299 ^dE^
**Cohesiveness**	1	0.65 ± 0.007 ^abC^	0.65 ± 0.005 ^abC^	0.64 ± 0.005 ^aD^	0.65 ± 0.006 ^abC^	0.64 ± 0.004 ^abD^	0.65 ± 0.005 ^bC^
	15	0.64 ± 0.004 ^bC^	0.63 ± 0.011 ^abB^	0.63 ± 0.007 ^aC^	0.63 ± 0.007 ^abB^	0.63 ± 0.007 ^abC^	0.63 ± 0.006 ^abB^
	30	0.63 ± 0.005 ^abB^	0.63 ± 0.005 ^bB^	0.62 ± 0.009 ^aBC^	0.63 ± 0.004 ^abB^	0.63 ± 0.006 ^abBC^	0.63 ± 0.007 ^bB^
	45	0.62 ± 0.007 ^abB^	0.61 ± 0.014 ^abA^	0.61 ± 0.010 ^aAB^	0.62 ± 0.007 ^abA^	0.62 ± 0.004 ^bB^	0.62 ± 0.005 ^abA^
	60	0.60 ± 0.011 ^aA^	0.61 ± 0.010 ^abA^	0.61 ± 0.009 ^abA^	0.61 ± 0.007 ^abA^	0.61 ± 0.008 ^abA^	0.62 ± 0.006 ^bA^
**Springiness (mm)**	1	4.36 ± 0.029 ^bA^	4.33 ± 0.020 ^abC^	4.34 ± 0.021 ^abC^	4.32 ± 0.031 ^abD^	4.31 ± 0.024 ^aB^	4.31 ± 0.039 ^aB^
	15	4.35 ± 0.020 ^dB^	4.26 ± 0.014 ^aB^	4.33 ± 0.021 ^cdC^	4.30 ± 0.024 ^bcCD^	4.29 ± 0.024 ^bAB^	4.29 ± 0.015 ^bAB^
	30	4.34 ± 0.018 ^cB^	4.24 ± 0.017 ^aB^	4.32 ± 0.019 ^cBC^	4.27 ± 0.027 ^abBC^	4.28 ± 0.023 ^abAB^	4.28 ± 0.025 ^bAB^
	45	4.26 ± 0.023 ^abCA^	4.23 ± 0.033 ^aB^	4.30 ± 0.019 ^cAB^	4.24 ± 0.036 ^abAB^	4.26 ± 0.029 ^abcA^	4.27 ± 0.031 ^bcAB^
	60	4.24 ± 0.024 ^bcA^	4.18 ± 0.031 ^aA^	4.28 ± 0.028 ^cA^	4.21 ± 0.053 ^abA^	4.25 ± 0.040 ^bcA^	4.26 ± 0.030 ^cA^
**Chewiness (NXmm)**	1	33.41 ± 1.25 ^aA^	34.15 ± 0.887 ^bA^	36.95 ± 0.543 ^bA^	37.95± 0.358 ^cA^	38.43 ± 0.990 ^cA^	38.55± 0.525 ^cA^
	15	34.64 ± 1.04 ^aA^	35.24 ± 0.792 ^aB^	37.96 ± 0.921 ^bA^	38.97 ±0.542 ^bcA^	39.29 ±0.831 ^cAB^	41.01 ± 0.752 ^dB^
	30	37.29 ± 1.04 ^bB^	35.88 ±0.984 ^abC^	42.00 ± 0.664 ^dB^	41.64 ± 0.877 ^dB^	40.14 ± 0.981 ^cB^	42.72 ± 0.771 ^dC^
	45	38.78 ± 0.84 ^bC^	36.23 ± 0.900 ^aC^	43.33± 0.928 ^cC^	42.63 ± 0.850 ^cB^	42.05 ± 0.870 ^cC^	43.29 ± 0.880 ^cC^
	60	41.26 ± 0.775 ^bD^	37.23 ± 0.839 ^aD^	45.98 ± 0.988 ^cD^	45.31 ± 0.652 ^cC^	45.15 ± 0.837 ^cD^	46.25 ± 0.904 ^cD^

Emulsified meat system with nanoemulsion (EMSN). The lowercase letters in the superscript indicate significant differences (*p* < 0.05) between treatments (rows). and uppercase letters indicate significant differences in each treatment with respect to time (columns) (*p* < 0.05).

**Table 4 foods-08-00357-t004:** Phenols, antioxidant activity via the DPPH and ABTS methods, and oxidative stability via the TBARS method in an emulsified meat system with nanoemulsion.

	Days	EMSN 0%	EMSN 1%	EMSN 2%	EMSN 3%	EMSN 4%	EMSN 5%
**Phenols mg GAE/100g**	1	^ND^	12.76 ± 0.345 ^aC^	13.29 ± 0.486 ^aC^	15.64 ± 0.177 ^bD^	19.86 ± 0.215 ^cD^	24.93 ± 0.170 ^dE^
	15	^ND^	12.22 ± 0.385 ^aC^	14.39 ± 0.049 ^bD^	14.47 ± 0.098 ^bC^	15.21 ± 0.098 ^cC^	18.09 ± 0.161 ^dD^
	30	^ND^	11.25 ± 0.098 ^aB^	12.31 ± 0.078 ^bB^	12.39 ± 0.345 ^bB^	15.10± 0.085 ^cC^	16.04 ± 0.085 ^dC^
	45	^ND^	10.45 ± 0.274 ^aA^	11.71± 0.148 ^aAB^	12.50 ± 0.098 ^bB^	13.25 ± 0.090 ^cB^	14.58 ± 0.085 ^dB^
	60	^ND^	10.40 ± 0.098 ^aA^	11.39 ± 0.098 ^bA^	11.56 ± 0.177 ^bA^	11.76 ± 0.085 ^bcA^	12.16 ± 0.177 ^cA^
**DPPH** **mg AAE/100g**	1	15.55 ±0.288 ^aC^	18.13 ± 0.377 ^bD^	19.20 ± 0.108 ^cD^	19.26 ± 0.188 ^cD^	19.76 ± 0.288 ^cD^	19.89 ± 0.288 ^cD^
	15	13.60 ± 0.188 ^aC^	17.88 ± 0.474 ^bD^	19.01 ± 0.474 ^cD^	18.69 ± 0.188 ^cD^	19.07±0.499 ^cdCD^	19.89 ± 0.288 ^dD^
	30	11.40 ± 0.474 ^aB^	16.05 ± 0.201 ^bC^	17.18 ± 0.343 ^cC^	17.94 ± 0.218 ^dC^	18.50 ± 0.288 ^deC^	18.94 ± 0.288 ^eC^
	45	11.77 ± 0.288 ^aB^	11.84 ± 0.108 ^abB^	12.34 ± 0.108 ^bB^	14.92 ± 0.188 ^cB^	15.30 ± 0.499 ^cB^	16.24 ± 0.188 ^dB^
	60	7.94 ± 0.188 ^aA^	10.14 ± 0.288 ^bA^	10.89± 0.288 ^bcA^	11.52 ± 0.499 ^cA^	12.53 ± 0.390 ^dA^	14.48 ± 0.288 ^eA^
**ABTS mg AAE/100g**	1	22.53 ± 0.492 ^aD^	28.98 ± 0.372 ^bC^	33.07 ± 0.492 ^cD^	33.93 ± 0.322 ^cC^	34.25± 0.445 ^cB^	37.59± 0.492 ^dC^
	15	20.92 ± 0.492 ^aC^	26.73 ± 0.222 ^bB^	29.85 ± 0.321 ^cC^	30.71 ± 0.234 ^cB^	32.32 ± 0.322 ^dA^	36.19 ± 0.322 ^eB^
	30	19.84± 0.492 ^aBC^	25.44 ± 0.492 ^bB^	28.66± 0.492 ^cBC^	29.74 ±0.322 ^cAB^	32.53 ± 0.492 ^dA^	35.01± 0.492 ^eAB^
	45	18.55 ± 0.492 ^aB^	25.65 ± 0.492 ^bB^	28.02± 0.186 ^cAB^	29.41 ± 0.322 ^dA^	31.78 ± 0.186 ^eA^	34.68 ± 0.492 ^fA^
	60	15.33 ± 0.201 ^aA^	23.29 ± 0.322 ^bA^	27.16 ± 0.322 ^cA^	29.20 ± 0.186 ^dA^	31.03 ± 0.234 ^eA^	34.36 ± 0.265 ^fA^
**TBARS mg MDA/Kg**	1	0.28 ± 0.006 ^fA^	0.26 ± 0.007 ^eA^	0.20 ± 0.001 ^dA^	0.11 ± 0.004 ^cA^	0.06 ± 0.006 ^bA^	0.04 ± 0.004 ^aA^
	15	0.36 ± 0.006 ^eB^	0.28 ± 0.008 ^dA^	0.22 ± 0.011 ^cA^	0.13 ± 0.006 ^bB^	0.09 ± 0.004 ^aB^	0.07 ± 0.006 ^aB^
	30	0.42 ± 0.011 ^fC^	0.31 ± 0.007 ^eB^	0.26 ± 0.004 ^dB^	0.20 ± 0.008 ^cC^	0.17 ± 0.003 ^bC^	0.12 ± 0.004 ^aC^
	45	0.58 ± 0.009 ^fD^	0.49 ± 0.010 ^eC^	0.41 ± 0.008 ^dC^	0.36 ± 0.003 ^cD^	0.29 ± 0.003 ^bD^	0.20 ± 0.005 ^aD^
	60	0.75 ± 0.007 ^fE^	0.57 ± 0.005 ^cD^	0.48 ± 0.008 ^dD^	0.38 ± 0.003 ^cE^	0.36 ± 0.003 ^bE^	0.27 ± 0.005 ^aE^

Emulsified meat system with nanoemulsion (EMSN), 2,2-diphenyl-1-picrylhydrazyl (DPPH), 2,2′-Azino-bis-3-ethylbenzothiazoline-6-sulfonic acid (ABTS), and 2-thiobarbituric acid reactive substances (TBARS), Not detected (ND), gallic acid equivalents (GAE), ascorbic acid equivalents (AAE) and malonaldehyde (MDA). The lowercase letters in the superscript indicate significant differences (*p* < 0.05) between treatments (rows), and uppercase letters indicate significant differences in each treatment with respect to time (columns) (*p* < 0.05).

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
