# Peer review of "Producing an Emulsified Meat System by Partially Substituting Pig Fat with Nanoemulsions that Contain Antioxidant Compounds: The Effect on Oxidative Stability, Nutritional Contribution, and Texture Profile"

_foods, 2019, doi:10.3390/foods8090357_

Round 1

Reviewer 1 Report

The manuscript  ID: FOODS -571025  described a W/O nanoemulsion formulated with orange essential oil and cactus acid fruit in substitution to pig fats. Authors reported  to obtained a EMS  with a good antioxidant capacity and  an improved nutritional contribution.

The manuscript English language should been revised and typos errors avoided, in addition there are mistakes in subject and in the relative verb.

1.Introduction

Line 44  report the Author name and the [ n° ] of the cited article, please correct the sentence.

Line 85  write the complete “AOAC” abbreviation.

Materials and Methods

2.1 In the preparation of the nanoemulsion Authors determined the droplet size by dymamic light scathering technique (line 68, 69) but in the  manuscript there aren’t report on droplet size and other physical properties. Authors are invited to inserted and discussed data of the nanoemulsion.

2.3 Line 75,  please correct the minced meat (1 cm2)

In the test the value of the used temperature are written  in a different way, please uniform  the typos ( for example  16° C, 72°c, 100°C etc..)

 2.5  This section described  only the total polyphenols determination  not the antioxidant activity , please correct the title.

2.6 and 2.7 described the two methods used to determine the antioxidant activity. In all the manuscript there is the ambiguous use of “antioxidant” or ”radical” words , please correct and uniform the  use of these terms

Line 103, line 110 and line 116 report  incomplete sentences

The  results of what ??? ,please insert the description to complete the sentence 

Line 108 and 114 please uniform the indication of     “Abs” or “absorbance” 

Line 123  6000g forces , please correct as 6000xg forces 

Results and Discussion

Table 3 . In the pdf version of the manuscript  the  units  on the vertical lines are not readable

In this section the increased value,  described by the Authors,  should be reported  as “about ….times/fold”

The manuscript  ID: FOODS -571025  described a W/O nanoemulsion formulated with orange essential oil and cactus acid fruit in substitution to pig fats. Authors reported  to obtained a EMS  with a good antioxidant capacity and  an improved nutritional contribution.

The manuscript English language should been revised and typos errors avoided, in addition there are mistakes in subject and in the relative verb.

1.Introduction

Line 44  report the Author name and the [ n° ] of the cited article, please correct the sentence.

Line 85  write the complete “AOAC” abbreviation.

Materials and Methods

2.1 In the preparation of the nanoemulsion Authors determined the droplet size by dymamic light scathering technique (line 68, 69) but in the  manuscript there aren’t report on droplet size and other physical properties. Authors are invited to inserted and discussed data of the nanoemulsion.

2.3 Line 75,  please correct the minced meat (1 cm2)

In the test the value of the used temperature are written  in a different way, please uniform  the typos ( for example  16° C, 72°c, 100°C etc..)

 2.5  This section described  only the total polyphenols determination  not the antioxidant activity , please correct the title.

2.6 and 2.7 described the two methods used to determine the antioxidant activity. In all the manuscript there is the ambiguous use of “antioxidant” or ”radical” words , please correct and uniform the  use of these terms

Line 103, line 110 and line 116 report  incomplete sentences

The  results of what ??? ,please insert the description to complete the sentence 

Line 108 and 114 please uniform the indication of     “Abs” or “absorbance”

Line 123  6000g forces , please correct as 6000xg forces

Results and Discussion

Table 3 . In the pdf version of the manuscript  the  units  on the vertical lines are not readable

In this section the increased value,  described by the Authors,  should be reported  as “about ….times/fold”

Reviewer 2 Report

The manuscript titled “Emulsified meat system by the partial substitution of pig fat with nanoemulsions containing bioactive compounds of cactus fruit and essential oil: oxidative stability, nutritional contribution and texture profile” from Isaac and co-workers presents work on the incorporation of the nanoemulsion in the emulsified meat system to improve its nutritional contribution. The topic is attractive and the data should be stimulating for many fields of interest. The work is relevant and includes sufficient novelty.

Therefore, please revise the following points.

The author should rewrite the title. It seems to be very big. Line 75, Change (1 cm2) to (1 cm2) The authors didn’t mention anything about nanoemulsion size or its characteristics, and these results should be included. It will be interesting to know the phenolic content, antioxidant activity of the nanoemulsion, could you please include it. In conclusion, the author should mention the best conditions of their findings.

I hope the authors may find my comments helpful. 

Round 2

Reviewer 2 Report

Since most of the comments are corrected by the author. The manuscript can be accepted as like that.